# Pervaporation Membranes for Seawater Desalination Based on Geo–rGO–TiO_2_ Nanocomposites: Part 2—Membranes Performances

**DOI:** 10.3390/membranes12111046

**Published:** 2022-10-26

**Authors:** Subaer Subaer, Hamzah Fansuri, Abdul Haris, Misdayanti Misdayanti, Imam Ramadhan, Teguh Wibawa, Yulprista Putri, Harlyenda Ismayanti, Agung Setiawan

**Affiliations:** 1Material Physics Laboratory, Physics Department, Universitas Negeri Makassar (UNM), Makassar 90223, Indonesia; 2Centre of Excellence on Green Materials & Technology (CeoGM-Tech), FMIPA, Universitas Negeri Makassar (UNM), Makassar 90223, Indonesia; 3Chemistry Department, Institut Teknologi Sepuluh Nopember (ITS), Kampus ITS Sukolilo, Surabaya 60111, Indonesia; 4Research Center for Mining Technology, National Research and Innovation Agency (BRIN), Building 820, Puspitek, Banten 15314, Indonesia

**Keywords:** desalination, geopolymer, membrane, pervaporation, permeate

## Abstract

This is part 2 of the research on pervaporation membranes for seawater desalination based on Geo–rGO–TiO_2_ nanocomposite. The quality of the Geo–rGO–TiO_2_ pervaporation membranes (PV), as well as the suitability of the built pervaporation system, is thoroughly discussed. The four membranes described in detail in the first article were tested for their capabilities using the parameters turbidity, salinity, total suspended solids (TSS), and electrical conductivity (EC). The membranes’ flux permeate was measured as a function of temperature, and salt rejection was calculated using the electrical conductivity values of the feed and permeate. Fourier-transform infrared (FTIR) and X-ray diffraction (XRD) techniques were used to investigate changes in the chemical composition and internal structure of the membranes after use in pervaporation systems. The morphology of the membrane’s surfaces was examined by means of scanning electron microscopy (SEM), and the elemental distribution was observed by using X-ray mapping and energy dispersive spectroscopy (EDS). The results showed that the pervaporation membrane of Geo–rGO–TiO_2_ (1, 3) achieved a permeate flux as high as 2.29 kg/m^2^·h with a salt rejection of around 91%. The results of the FTIR and XRD measurements did not show any changes in the functional group and chemical compositions of the membrane after the pervaporation process took place. Long-term pressure and temperature feed cause significant cracking in geopolymer and Geo–TiO_2_ (3) membranes. SEM results revealed that the surface of all membranes is leached out, and elemental distribution based on X-ray mapping and EDS observations revealed the addition of Na+ ions on the membrane surface. The study’s findings pave the way for more research and development of geopolymers as the basic material for inorganic membranes, particularly with the addition of rGO–TiO_2_ nanocomposites.

## 1. Introduction

In recent decades, the selection of membrane materials and technology for seawater desalination has advanced rapidly, driven by the growing need to ensure the availability of clean water around the world. Membrane materials are becoming more diverse and extensive, ranging from organic polymers to inorganic materials and mixed-matrix membranes to novel 2D materials, such as graphene [1]. Membrane separation technology, which includes reverse osmosis (RO), membrane distillation (MD), and pervaporation (PV), is becoming increasingly popular for seawater desalination due to its high efficiency, low energy and chemical consumption, and ease of operation [2,3].

The membrane used in pervaporative desalination is specifically designed to be hydrophilic or to be a molecular sieving porous membrane that selectively allows water molecules to pass through. The chemical potential gradient between the feed and permeate sides of the membrane drives permeate mass transfer in PV. After the permeate water vapor condenses on the other side of the membrane, the purified water is collected. Figure 1 illustrates a simple principle of the PV membrane in separating the feed constituent into the desired permeate [4]. Fouling and wetting are two main flaws in membrane technology that must be addressed. These two variables were found to have little effect on the PV membrane. Because the hydrophilic membrane in PV does not evaporate volatile organic compound permeation, the permeate water is organic-free [5,6,7]. 

PV membranes can operate at low temperatures ranging from 40 to 80 °C. Despite this, due to their low flux, PV membranes have not been widely used for seawater desalination. To increase permeate flux, the membrane must have a high affinity and a specific molecular structure that allows for mass transport [7,8]. To meet the aforementioned requirements, researchers are focusing on the materials and performance of PV membranes, which range from organic polymers, such as PE and cellulose, to microporous inorganic membranes made from zeolite–amorphous silica–based membranes [1,7,9,10]. Materials based on hybrid organic–inorganic composites, ceramics, and carbon-based materials, such as graphene oxide (GO), reduced graphene oxide (rGO), and graphene (G) forming thin-film composites (TFC), have made remarkable progress as novel and high-performing PV membranes [6,11,12]. 

The fabrication of a thin-film composite (TFC), which is composed of m-phenylenediamine (MPD) and trimesoyl chloride (TMC) on a porous substrate, has been the primary method for manufacturing reverse osmosis (RO) membranes. The TFC RO membrane exhibits remarkable performance in producing pure water from seawater with high flux and salt rejection and good mechanical strength. However, it has been discovered that TFC RO membranes are prone to fouling and influence the cost of desalination. The development of thin-film nanocomposite (TFN) RO membranes is one concept that might be used to address the drawbacks of TFC RO. This form of membrane combines high hydrophilicity, antifouling, and antimicrobial qualities with the functional properties of nanoparticles [13,14]. Currently, another alternative to enhance the performance of nanocomposite membranes is to fabricate membranes using the layer-by-layer (LBL) method by integrating different kinds of nanomaterials with polymer polyelectrolytes [15]. 

In recent years, a number of studies have reported the incorporation of nanomaterial into a polymer to fabricate nanocomposite PV membranes for seawater desalination, such as graphene oxide (GO) [16], chitosan/GO [6], AgNPs [17], rGO–TiO_2_ [18], Al_2_O_3_ NPs [19]. The next generation of high-performance membranes for PV desalination could be developed using nanocomposite membranes, which have the potential to compete with established RO membranes and overcome the shortcomings of low permeability and salt rejection, improved hydrophilicity and mechanical strength, and enhanced antifouling and antibacterial properties. 

Geopolymer is an inorganic polymer derived from aluminosilicate minerals that have progressed from a simple binder for concrete and structural materials to advanced materials with diverse application potentials, such as high-quality composites, self-cleaning and antibacterial materials, and supercapacitors [2,13,14]. Recently, geopolymer has gained popularity as a filter or membrane for sieving various heavy metals, wastewater treatment into clean water [15,16], and seawater desalination [17,18]. 

He et al. [20] reported on the development of geopolymer-based material for seawater desalination. The Al_2_O_3_–2SiO_2_–Na_2_O geopolymer membrane used by the authors was synthesized by a gel casting procedure, and it was afterwards hydrothermally converted into a NaA zeolite. The generated membrane, which has a thickness of 10 mm and a feed temperature of 90 °C, achieves a flux of 3.86 kg/m^2^·h and a salt rejection of about 99%. 

A zeolitized geopolymer-based pervaporation membrane was developed and reported recently by Abukhadra et al. [21]. Natural diatomite and kaolinite were used to synthesize the geopolymer at room temperature. A zeolitized geopolymer membrane was then generated by hydrothermal zeolite growth at 100 °C for 24 h. The authors examined the influence of the membrane thickness and feed temperature on the flux and salt rejection capacity of the membrane. Their results showed that for the tested membrane at a thickness of 3 mm and a feed temperature of 90 °C, the water flux reached values of 7.05 kg/m^2^·h with a salt rejection of around 99.57% after 130 h of pervaporation time. 

The current study aims to investigate the exceptional properties and performance of Geo–rGO–TiO_2_NPs as PV membranes for seawater desalination. The first part of this article [22] publicized various features and properties of the Geo–rGO–TiO_2_ NP membrane, and in this part 2, the performance of the membranes, as well as the condition of the pervaporation system for seawater desalination based on the experimental results, is discussed in detail. Several shortcomings and future prospects for the development of Geo–rGO–TiO_2_NPs as PV membranes are also discussed. The findings of this study are expected to be a significant contribution to the advancement of geopolymer as a novel material for PV inorganic membranes.

## 2. Materials and Methods

The membrane geopolymer–rGO–TiO_2_NPs used in this study were produced from metakaolin, TiO_2_NPs, and rGO. Metakaolin is produced through the dehydroxilation of kaolin at 750 °C for 4 h. Kaolin was purchased from Pt. Intraco, Makassar, Indonesia. Graphene oxide was synthesized by using graphite, which was purchased from Pt Pelinda Sarana Sukses, Jakarta, Indonesia. The properties of the raw materials, as well as the synthesis method of geopolymer, rGO, and the membranes, followed by their microstructure properties, distribution of porosity, mechanical strength, and hydrophilic properties, have been described in the previous article [22]. 

Geopolymer paste was made from metakaolin and activated with an alkaline solution (NaOH + Na_2_OSiO_2_ + H_2_O), then sealed in a closed mold and cured at 70 °C for 2 h. The TiO_2_NPs used in this study were supplied by Sanno, Jakarta, Indonesia, and had an average particle size of 20 nm. Hummer’s modified method is used to synthesize rGO nanosheets from graphite. The synthesis of geopolymer and geopolymer–rGO–TiO_2_ composites is carried out manually. Table 1 shows the four types of pervaporation membrane compositions developed in this study for seawater desalination. Each composition is made up of several specimens for mechanical testing, microstructure examinations, and pervaporation experiments. The amount of geopolymer paste in each membrane is the same. The variations of rGO addition are intended to improve the photocatalytic properties of TiO_2_NPs while increasing the salt rejection capacity of the formed composite.

Figure 1 illustrates the schematic production of geopolymer–rGO–TiO_2_NPs to be used as pervaporation membranes for seawater desalination. The effective area of the membrane is 39.81 cm^2^.

In a pervaporation system, the flux of the permeate (*J*, kg/m^2^·h) was calculated by using the formula:(1)$$J=\frac{m}{A\times \Delta t}$$
where *m* is the mass of the permeate (kg), *A* is the effective area of the membrane (m^2^), and Δ*t* is pervaporation time (h).

The salt rejection (*R*, %) was calculated by using the equation:(2)$$R=\frac{Sf-Sp}{Sf}\times 100\%$$
where *Sf* and *Sp* are the salt concentration in the feed and permeate, respectively. The salt concentrations in the feed and permeate were determined through conductivity measurement. The seawater used in this study as a feed was obtained from the Jeneponto District, a salt-producing area in South Sulawesi Province, Indonesia. This choice was made due to the high salinity and brightness, and the possibility of contamination with impurity materials is very low.

For the purposes of this study, four membranes were produced, as shown in Figure 2, namely, geopolymer, Geo–TiO_2_ (3), Geo–rGO–TiO_2_ (0.5, 3), and Geo–TiO_2_ (1, 3). The resulting membranes were stored at room temperature for 28 days before the pervaporation process was carried out.

The pervaporation system for seawater desalination developed in this study was based on a system described by Qian et al. [6]. However, the membrane design, as well as the membrane house, was distinct. The pervaporation design depicted in Figure 3a consists of a 2000 mL flask to house the seawater (feed). The feed was pumped into the membrane via PVC lines by a Shimizu Model PS-116 water pump connected to a DSK-8 automatic pressure control. The feed was heated up to 80 °C using an electronic hot plate of the type Sitiantek ST-946C. A thermocouple was placed inside the flask to measure the feed temperature and connected to a Constant TC 12 digital thermometer (supplied by Karya Mandiri Techindo, Jakarta, Indonesia). The pervaporation was carried out at three different feed temperatures: 40, 60, and 80 °C. The membrane was placed inside a transparent membrane house (50-150G RO membrane housing) (purchased from HID Membrane Co., LTD, China), where the retentate (seawater) and permeate (freshwater) were separated. The permeate was drawn into the water-cooling Graham condenser by a Pfeiffer vacuum pump, UNO-30 (supplied by Mack Vacuum Technologies, Inc., Longwood, USA), and collected in a glass container sitting on the surface of a Joil D1/D8 digital balance for measuring the permeate mass. Figure 3b shows a photograph of the pervaporation system used in the study. The system operates at a maximum water pressure of approximately 2 psi and the best vacuum condition of approximately 0.75 bar.

The quality of the permeate as a result of pervaporation is determined using the parameters pH, turbidity, total suspended solid (TSS), salinity, and electrical conductivity (EC). The test was carried out at the Environmental Management Agency in South Sulawesi Province, Indonesia.

The physicochemical properties after the membrane work in the seawater desalination pervaporation system were tested with FTIR, XRD, and SEM–EDS techniques. FTIR was employed to examine the functional groups of the membranes, XRD was used to observe the crystallinity and chemical compositions of the membranes, and SEM–EDS was performed to study the microstructure and the elemental composition of the membranes.

## 3. Results and Discussion

### 3.1. The Membrane Performance and Desalination Results

The flux permeate and membrane durability were tested in preliminary experiments at different pressures and temperatures, vacuum pressure, water cooling temperature, and pervaporation time. Following two trial experiments, it was discovered that the geopolymer and Geo–TiO_2_NPs membranes exhibited significant cracking. Based on these findings, the pervaporation for all prepared membranes was conducted by adjusting the feed pressure to 2 psi and setting the feed temperatures to 40, 60, and 80 °C; the vacuum pressure to 0.75 bar, and to water cooling temperature to 15 °C. Each series of tests required about 5 to 6 h per day, and it continued the following day until there was enough permeate for testing. Under the established pervaporation parameters, it was discovered that geopolymers and Geo–TiO_2_ (3) membranes still had significant cracks and thus revealed relatively high flux permeate, whereas Geo–rGO–TiO_2_ (0.5, 3) and Geo–rGO–TiO_2_ (1, 3) membranes had no surface cracks. The presence of rGO appears to give the two membranes greater mechanical and thermal resistance than geopolymer and geo–TiO_2_ (3) membranes. The enhancement of microstructural properties, mechanical strength, and thermal resistance of geopolymer through the addition of rGO or GO has also been reported by Shu et al. [23] and Li et al. [24]. 

Based on Figure 4, the geopolymer and Geo–TiO_2_ (3) membranes developed substantial cracks with crack widths ranging from 0.07 ± 0.01 to 0.17 ± 0.01 mm after the entire pervaporation series was completed. The cracks could have been caused by fairly high pressure and temperature feed that hit the membrane’s surface for a prolonged period of time. Cracks on the surface render the membranes inoperable. However, no visible cracks were found on the surface of the Geo–rGO–TiO_2_ membranes. The strength of all membranes measured by splitting tensile was between 0.27 and 0.35 MPa, indicating that they should be able to withstand the high pressure of the feed [22]. The measurements, however, were carried out at room temperature. As the temperature of the feed increases, the strength of the membrane decreases, causing crack propagation, as seen in geopolymer and Geo–TiO_2_ (3) membranes.

To determine the membranes’ long-term durability, the pervaporation time required to produce a 1 L volume of permeate was used as the standard. The Environmental Management Agency in South Sulawesi Province, Indonesia, requires this volume of permeate to measure turbidity, total suspended solids (TSS), salinity, and electrical conductivity (EC). Table 2 shows the pervaporation time to produce 1 L permeate. 

It is worth noting that the membranes Geo–rGO–TiO_2_ (0.5, 3) and Geo–rGO–TiO_2_ (1,3) can operate for over 100 h without visible damage. Permeate flux is high in the geopolymer and Geo–TiO_2_ (3) membranes, requiring less time to produce 1 L of permeate because both membranes experience cracks during the pervaporation process.

Figure 5 depicts the flux permeate of the membranes at different temperatures. The geopolymer and Geo–TiO_2_ (3) membranes have the highest flux at all feed temperatures, which is known to be due to the formation of cracks on the membrane’s surface.

Table 3 shows the feed and permeate parameters, such as turbidity, salinity, total suspended solid (TSS), electrical conductivity (EC), and flux permeate (J) of each membrane.

Table 3 shows that the salt rejection for the Geo–rGO–TiO_2_ (0.5, 3) and Geo–rGO–TiO_2_ (1,3) membranes were around 73.01% and 91.26%, respectively. Although the salt rejection value of the two membranes did not reach 99%, the addition of rGO–TiO_2_NPs to the matrix geopolymer appears to play a significant role in separating NaCl from H_2_O. It is highly probable that the low salt rejection of Geo–rGO–TiO_2_ (0.5, 3) and Geo–rGO–TiO_2_ (1, 3) membranes is due to the average size of the pore, which is 12.90 and 13.40 nm, respectively. In contrast to the NaA zeolite geopolymer membrane, which has an average pore size of about 3.77 nm and achieves 99% salt rejection [20], the Geo–rGO–TiO_2_ membranes generated in this study have substantially larger average pore sizes. 

The turbidity value of the permeate for all membranes is smaller than 5 NTU, which means that it meets the maximum turbidity criteria set by WHO [25]. The measured TSS values of the membrane geopolymer and Geo–TiO_2_ (3) appear to be very high, exceeding the TSS value of seawater (feed). This value is unreasonable and is most likely due to the presence of alkali carbonate crystals detached from the surface and the crack area of geopolymers and the Geo–TiO_2_ (3) membranes. One of the concerns with geopolymers made from metakaolin or fly ash is the formation of white needle-like alkali carbonate crystals known as efflorescence, which is caused by the mobility of excess alkali when the geopolymer is exposed to open air or submerged in water for an extended period of time [26,27]. Despite the absence of surface cracks in the Geo–rGO–TiO_2_ (0.5,1) and Geo–rGO–TiO_2_ (1,5) membranes, NTU and TSS values are still high and detectable when NaCl rejection reaches 91%. Both membranes could also experience efflorescence, and in this case, the presence of high NTU and TSS values is due to the formation of alkaline carbonate crystals. The efflorescence in the geopolymer made from metakaolin can be controlled by adjusting the molar ratio of Na_2_O/Al_2_O_3_ of the alkali solution used [28], or the addition of soluble silicate, silica fume, or silicone oil [29]. It is unclear whether the addition of rGO will be able to control the formation of efflorescence on the surface of the geopolymer. More research is needed to confirm this.

Table 4 compares the membrane performance produced in this study with that of other inorganic membranes reported by others. It is clear that the Geo–rGO–TiO_2_ membranes require further optimization in order to achieve comparable performance with other membranes, particularly those made of geopolymer materials.

### 3.2. FTIR Characterization

Figure 6 shows the FTIR spectrum of the membranes after the pervaporation process was completed. The stretching and bending vibrations of the functional groups of (OH)^−^ for the membrane geopolymer are centered on the broad band at approximately 3448 and 1644 cm^−1^, respectively. The centers of these broad bands are slightly shifted in other membranes, with the largest shift occurring in the Geo–rGO–TiO_2_ (1, 3) membrane, specifically at positions 3464 and 1649 cm^−1^. The bands that are identified as the primary functional groups of geopolymers are as follows: The asymmetrical stretching vibration of the Si–O–Si bond is observed at around 1014 cm^−1^, while the bending vibration of the Si–O–Si bond is discovered at around 456 cm^−1^. The band around 861 cm^−1^ is attributed to the stretching vibration of the Si–O bond, and the stretching vibration of the Al–O–Si bond is approximately 551 cm^−1^. The bending vibration of Al(IV) –O–Si in the cyclic structure of the geopolymer is found to be 687 cm^−1^ [31,32,33]. Except for the asymmetrical stretching vibration of the Si–O–Si bond, the addition of TiO_2_NPs and rGO to the geopolymer network causes a noticeable peak shift in the other primary groups. The functional group of the Ti–O–Ti bonds in the membrane to which TiO_2_NPs are added vibrates at a wave number of approximately 1403 cm^−1^ [34]. The asymmetric stretching of the O–C–O bond within the Na_2_CO_3_ group is centered around 1412 cm^−1^ [35]. 

### 3.3. XRD Characterization

Figure 7a shows XRD patterns of all membranes after the entire series of pervaporation experiments is completed. The phases of quartz and α-SiO_2_ originate from kaolin and remain in the form of crystalline after the dehydroxylation process at 750 °C. The amorphous hump in the metakaolin-based geopolymer between 20°–30° 2θ shift slightly to the lower 2θ in the Geo–TiO_2_ and Geo–rGO–TiO_2_ samples. This is due to the reorganization of geopolymer networks during the addition of rGO and TiO_2_ [36]. The addition of rGO and TiO_2_NPs forms the semicrystalline network of the geopolymer, indicating the formation of nanocomposite material. The crystalline phase of NaAl_2_(AlSi_3_)O_11_ is present in all samples, but its amount decreases significantly as rGO and TiO_2_NPs are added to the geopolymer paste (Figure 7b). The reflection peak of TiO_2_NPs with a crystallographic plane of (101) is at 2θ = 25.57° and d-spacing = 3.4807 angstroms (PDF#01-075-2552), while the reflection peak of rGO with a crystallographic plane (002) appears at 2θ = 26.00° with d-spacing = 3.424 angstroms (PDF#01-077-7164). It appears that the reflection angle of TiO_2_NPs overlaps with the reflection angle of rGO. The XRD characterization of the samples does not reveal a NaCl phase after pervaporation is conducted.

### 3.4. SEM–EDS Examinations

Morphological and phase changes in the surface and cross-section membranes after the pervaporation process are examined using SEM. The EDS is employed to determine the elemental composition of the membrane surface. Biofouling by various microorganisms and polymeric substances as well as Na^+^ and Cl^−^ ion deposition on membrane surfaces is a major issue in membrane performances made of organic polymers or inorganic membranes [17,37]. Figure 8 shows SEM images of the surfaces of all membranes after pervaporation at different magnifications. The entire surface of the membrane undergoes morphological destruction as a result of the high temperature and feed pressure, resulting in a significant amount of geopolymer paste being peeled off, as can be seen at higher magnifications of SEM images. At higher magnifications, the presence of TiO_2_NPs and the rGO sheet is more visible in the geopolymer matrix.

Geo–rGO–TiO_2_ membrane SEM images reveal the presence of microscopic cracks that are not visible to the naked eye. The crack could be a secondary crack caused by internal stress in the SEM chamber. Unlike the geopolymer and Geo–TiO_2_ (3) membranes, the cracks formed are visible to the naked eye and quite long and wide, as illustrated in Figure 4. It was found that the presence of rGO–TiO_2_ nanocomposite appears to increase the mechanical strength of the membrane and is capable of preventing cracks caused by high pressure and feed temperature for an extended period of time. Previous research [18,38,39] demonstrated that the addition of rGO improved the microstructure and increased the mechanical strength of the geopolymer.

The distribution of elemental components on the surfaces of geopolymer and Geo–rGO–TiO_2_NPs nanocomposites after pervaporation is revealed by X-ray mapping, as shown in Figure 9. The images were taken at a magnification of 1000× and HV = 20 kV.

The X-ray mapping results show that the elemental compositions are well distributed on the surface of the membranes as expected, and no impurity elements indicative of biofouling are found. These findings demonstrate that geopolymers are materials that are resistant to seawater attacks [40,41]. The addition of rGO as a 2D material acts as a nanochannel capable of passing H_2_O molecules and retaining Na^+^ and Cl^−^ ions, while TiO_2_NPs as photocatalyst materials act as a biofouling material. Furthermore, TiO_2_NPs are layered with rGO or graphene to increase the specific surface area and charge carrier mobility and thus the photocatalytic efficiency [13,18,38]. The quantitative analysis of elemental compositions based on EDS results is shown in Figure 10. The percentage of Na on the surface of all membranes appears to be greater than the initial Na composition during membrane synthesis. This shows that during pervaporation, Na^+^ ions are successfully separated from the feed and deposited on the surface or bulk membrane.

## 4. Conclusions

Geo–rGO–TiO_2_ nanocomposites are cutting-edge inorganic materials that could be used as pervaporation membranes. The membrane with a thickness of around 2 mm showed a permeate flux of 2.29 kg/m^2^·h and a salt rejection of 91% at a pervaporation temperature of 60 °C. The membrane does not sustain surface damage during the pervaporation process and does not exhibit biofouling after prolonged exposure to seawater. Pervaporation membranes made of pure geopolymer materials and geopolymer–TiO_2_NP composites have salt rejection rates of more than 50% and 60%, respectively, but both exhibit significant cracking after pervaporation. These findings suggest that the presence of rGO in the matrix geopolymer along with TiO_2_NPs increases the physical and thermal resistance of the membrane in addition to increasing membrane selectivity. It was discovered that the quality of the water pump used was inadequate for the pervaporation of seawater desalination because it was prone to rusting by soiling pipes and membranes. Increased permeate flux and salt rejection are open research areas, allowing Geo–rGO–TiO_2_ membranes to compete with other inorganic membranes. The results of this study provide great hope for the development of geopolymer-based pervaporation membranes through the addition of materials, such as TiO_2_NPs and graphene-based materials.

## Figures and Tables

**Figure 1 membranes-12-01046-f001:**
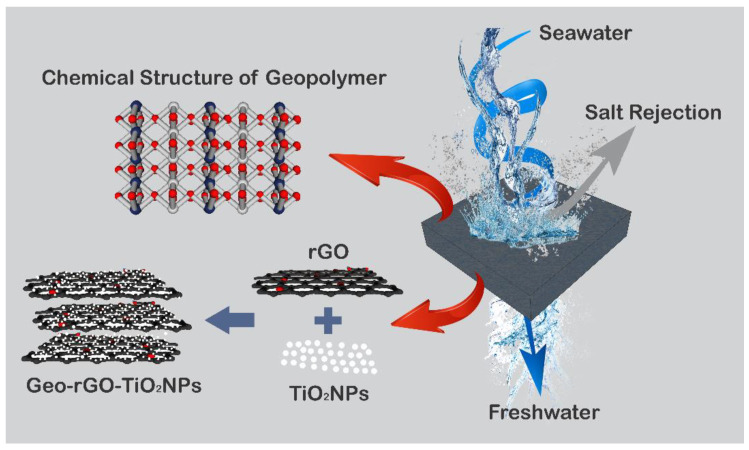
Schematic production of the geopolymer–rGO–TiO_2_NPs as a pervaporation membrane.

**Figure 2 membranes-12-01046-f002:**
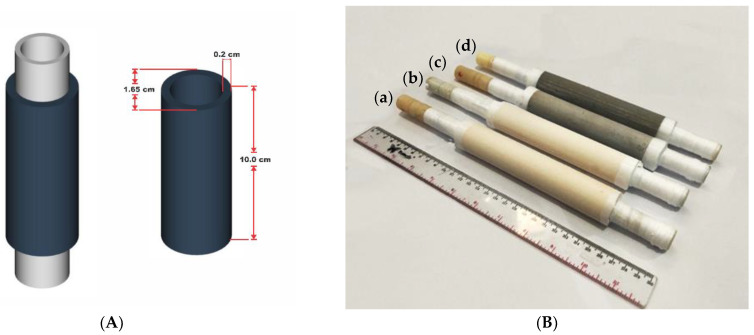
(**A**) Membrane design with ceramic support and (**B**) PV membranes produced in this study: (**a**) geopolymer, (**b**) Geo–TiO_2_ (3), (**c**) Geo–rGO–TiO_2_ (0.5, 3), and (**d**) Geo–rGO–TiO_2_ (1, 3).

**Figure 3 membranes-12-01046-f003:**
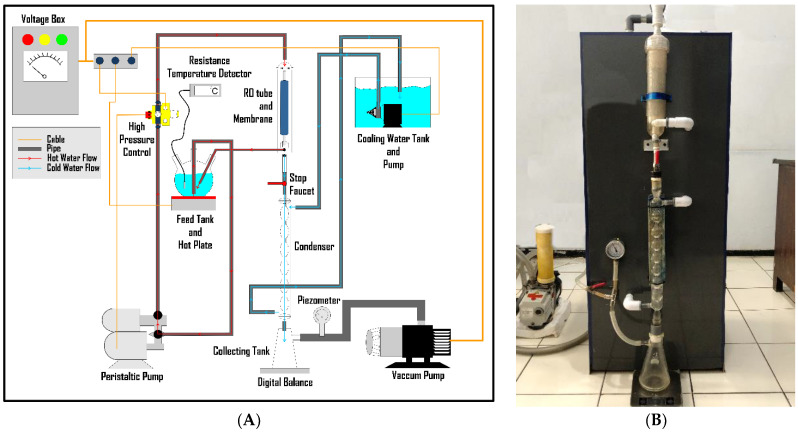
(**A**) Design of the pervaporation system built for this study and (**B**) photograph of the pervaporation system in working condition.

**Figure 4 membranes-12-01046-f004:**
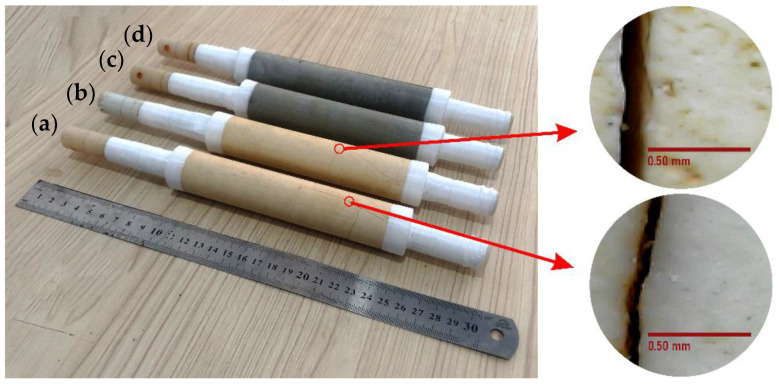
A photograph of the membranes ((**a**) geopolymer, (**b**) geo–TiO_2_ (3), (**c**) Geo–rGO–TiO_2_ (0.5, 3), (**d**) Geo–rGO–TiO_2_ (1, 3)) after the entire series of experiments was completed. A digital optical microscope was used to examine cracks on the surface of the membranes.

**Figure 5 membranes-12-01046-f005:**
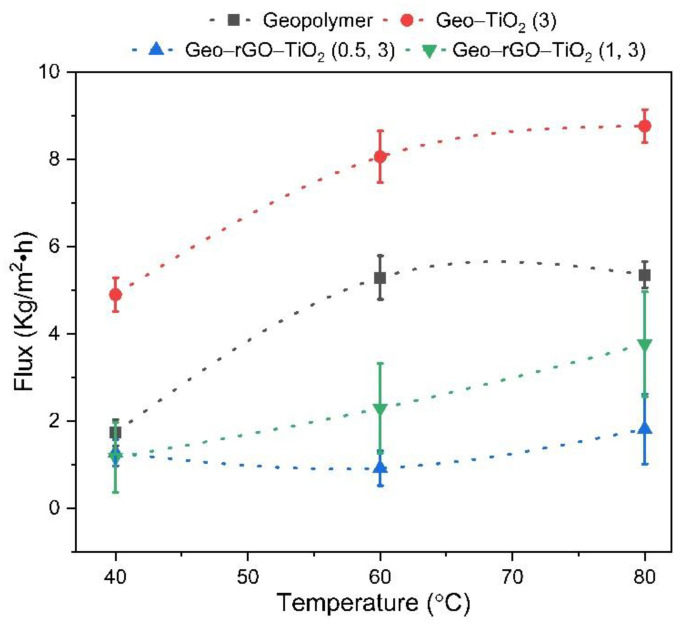
The permeate flux (J) as a function of feed temperature (T) for four different membranes. The error bars represent the standard deviation of three measurements.

**Figure 6 membranes-12-01046-f006:**
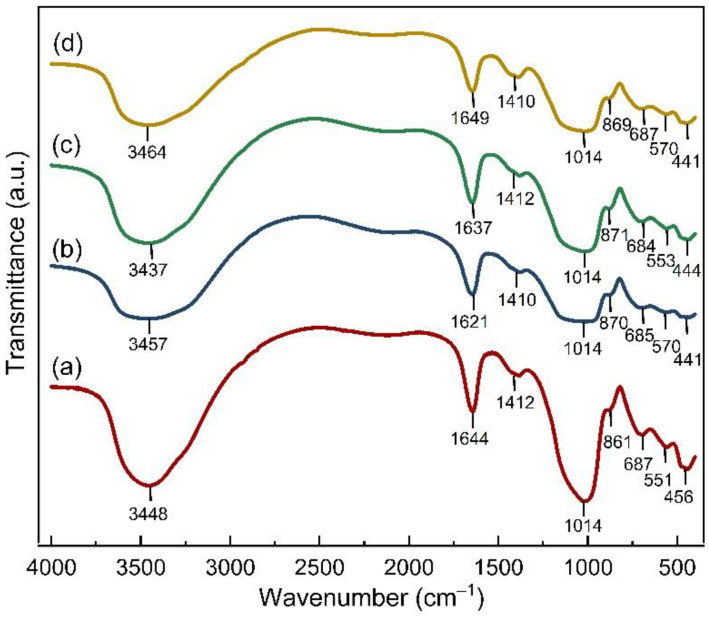
The FTIR spectrum of the membranes after a series of pervaporation experiments indicating the vibrational bands of the functional groups of (**a**) geopolymer, (**b**) Geo–TiO_2_ (3), (**c**) Geo–rGO–TiO_2_ (0.5, 3), and (**d**) Geo–rGO–TiO_2_ (1, 3).

**Figure 7 membranes-12-01046-f007:**
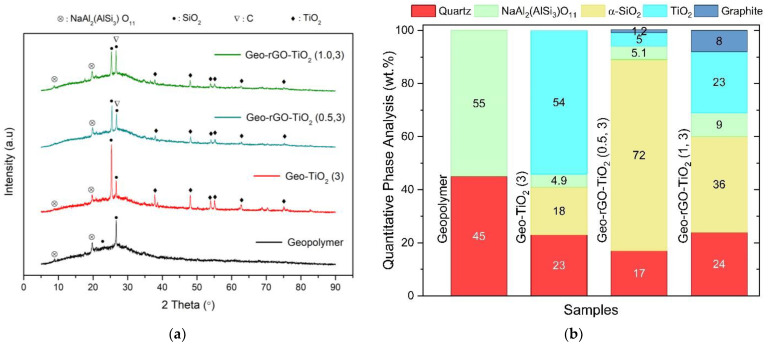
(**a**) XRD patterns and (**b**) quantitative phase distribution of the membranes: geopolymer, Geo–TiO_2_ (3), Geo–rGO–TiO_2_ (0.5, 3), and Geo–rGO–TiO_2_ (1.3) after pervaporation.

**Figure 8 membranes-12-01046-f008:**
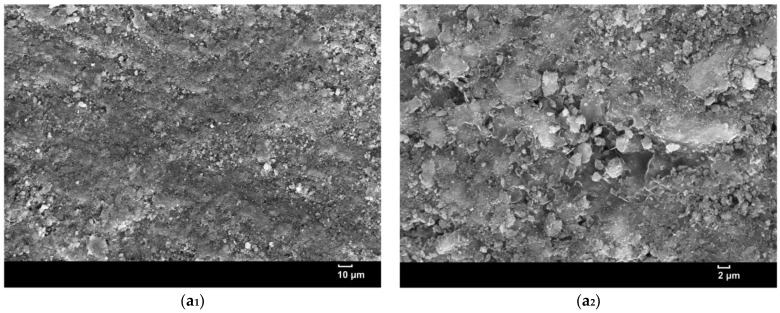
SEM images of membrane surfaces after pervaporation. (**a_1_**,**a_2_**) Geopolymer, (**b_1_**,**b_2_**) Geo–TiO_2_ (3), (**c_1_**,**c_2_**) Geo–rGO–TiO_2_ (0.5, 3), and (**d_1_**,**d_2_**) Geo–rGO–TiO_2_ (1, 3). The phase locations of TiO_2_NPs, rGO, and cracks are depicted in the figure.

**Figure 9 membranes-12-01046-f009:**
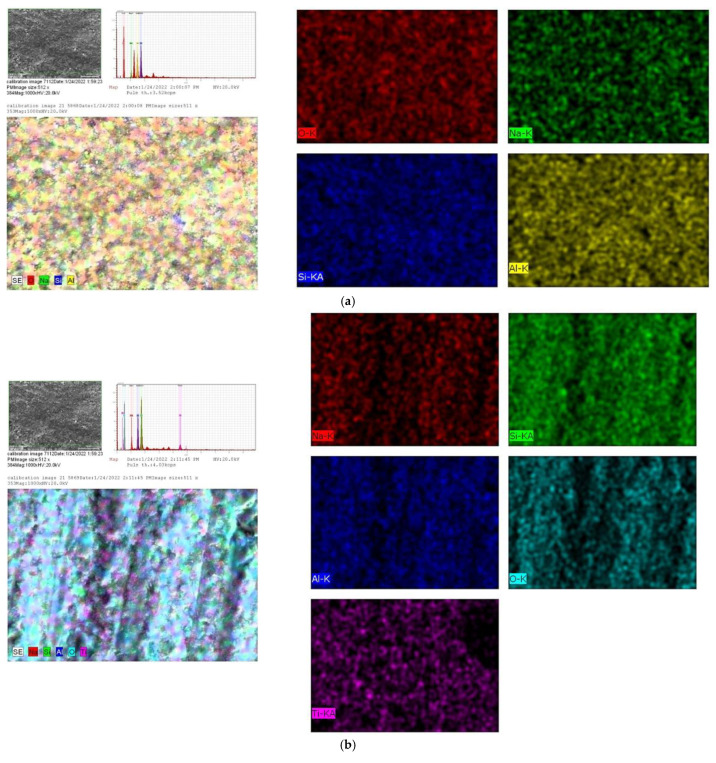
X-ray mapping showing the elemental distribution on the surface of the membranes (**a**) geopolymer, (**b**) Geo–TiO_2_ (3), (**c**) Geo–rGO–TiO_2_ (0.5, 3), and (**d**) Geo–rGO–TiO_2_ (1, 3).

**Figure 10 membranes-12-01046-f010:**
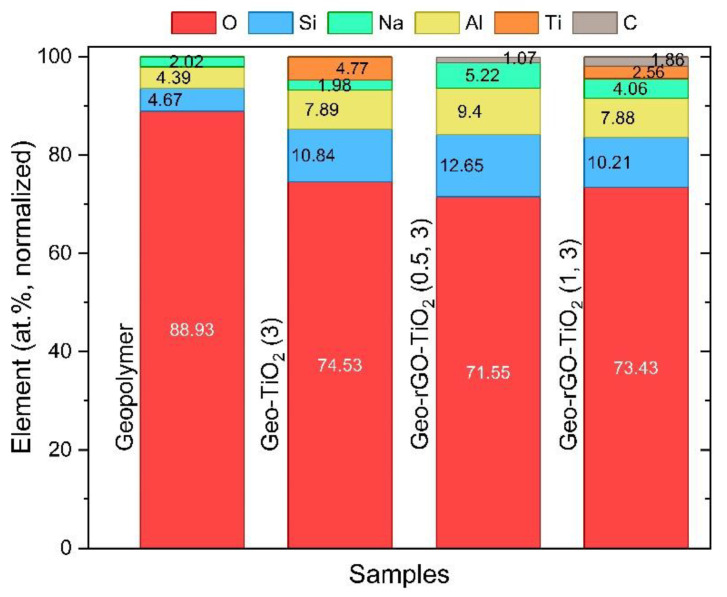
The distribution of elements on the surface of the membranes after pervaporation. The high percentage of Na indicates the presence of a Na deposit from the feed during the pervaporation.

**Table 1 membranes-12-01046-t001:** The composition of starting materials (in grams) of the pervaporation membranes produced in this study.

Sample	Metakaolin	NaOH	Na_2_OSiO_3_	H_2_O	rGO	TiO_2_
Geopolymer	30	3	24	9	0	0
Geo–TiO_2_ (3)	30	3	24	9	0	3
Geo–rGO–TiO_2_ (0.5, 3)	30	3	24	9	0.5	3
Geo–rGO–TiO_2_ (1, 3)	30	3	24	9	1.0	3

**Table 2 membranes-12-01046-t002:** Pervaporation time required to produce 1 L of permeate.

Membrane	Flux (kg/m^2^·h)	Pervaporation Time (h)
Geopolymer	5.29	47.5
Geo–TiO_2_ (3)	8.06	31.2
Geo–rGO–TiO_2_ (0.5, 3)	0.92	273
Geo–rGO–TiO_2_ (1, 3)	2.29	109

**Table 3 membranes-12-01046-t003:** Seawater parameters and pervaporation results of four different membranes.

Seawater/Permeate of the Membranes	Flux (kg/m^2^·h)	Turbidity (NTU)	Salinity (ppt)	TSS (mg/L)	EC (mS/cm)	Rejection (%)	pH
Seawater	-	26.2 ± 0.26	30.25 ± 0.30	25.0 ± 0.50	38.2 ± 2.29	-	7.12 ± 0.01
Geopolymer	5.29 ± 0.50	0.14 ± 0.01	9.72 ± 0.10	37.5 ± 0.75	18.30 ± 1.10	52.09 ± 1.04	7.72 ± 0.01
Geo–TiO_2_ (3)	8.06 ± 0.59	0.82 ± 0.02	7.77 ± 0.08	31.5 ± 0.63	14.68 ± 0.88	63.32 ± 1.27	7.60 ± 0.01
Geo–rGO–TiO_2_(0.5, 1)	0.92 ± 0.40	0.82 ± 0.02	6.81 ± 0.07	12.5 ± 0.25	10.31 ± 0.62	73.01 ± 1.46	7.71 ± 0.02
Geo–rGO–TiO_2_ (1, 3)	2.29 ± 1.03	1.52 ± 0.03	2.49 ± 0.02	15 ± 0.30	3.34 ± 0.20	91.26 ± 1.83	7.60 ± 0.02

**Table 4 membranes-12-01046-t004:** Comparison between the produced membrane performance and other inorganic membranes.

Membrane	Thickness (mm)	Feed Temperature (°C)	Flux (kg/m^2^·h)	Rejection (%)	References
NaZ zeolite–geopolymer	10	90	3.86	99.5	[20]
Zeolitized diatomite/geopolymer	3	90	7.63	97.6	[21]
Hydroxy sodalite	2	202	3.5	99.99	[30]
Geo–rGO–TiO_2_ (1, 3)	2	60	2.92	91.26	This study

## Data Availability

Not applicable.

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
