# Peer review of "Pervaporation Membranes for Seawater Desalination Based on Geo–rGO–TiO2 Nanocomposites: Part 2—Membranes Performances"

_membranes, 2022, doi:10.3390/membranes12111046_

Round 1

Reviewer 1 Report

This manuscript, written by Subaer et al., is a continuing study based on the previous work (Membranes 11.12 (2021): 966). In this work, the authors investigated the performance of four different membranes in purifying seawater. In general, this work is meaningful, as it is a downstream demonstrator of the authors’ first work. However, before being accepted, I have several major concerns need to be addressed. Especially, the sample performances were not measured at the same level, which significantly compromised your data credibility.

1.     In materials and methods, could you please briefly introduce your synthesis process? If I don’t read your first work, I don’t understand what these membranes are. For example, the difference between Geo-rGO-TiO2 (0.5, 3) and Geo-rGO-TiO2 (1, 3).

2.     Line 145 – 158. First of all, if the experimental sample was physically affected by experimental conditions, then, you need to optimize the experimental conditions. Otherwise, your data was significantly compromised. For example, due to membrane cracking, you reduced the pressure. What about other membranes? Did you also reduce the pressure for other sets of experiments? Second of all, although you reduced the pressure, you observed membrane cracking in the end. Except for the variables (e.g., flux) you were interested in, you brought additional variables (i.e., materials strength) in the experiments, which invalidated your data in Figure 5. In this case, I could conclude the higher flux for Geopolymer and Geo-TiO2 was due to membrane cracking. Please optimize your experimental conditions to ensure all samples were measured at the same level.

3.     Line 145 – 158, please reorganize your data set. After describing Figure 5, you may be followed by Figure 6 and so on. Also, can you please provide a figure or table to compare the flux from this work with others? Or a brief description.

4.     Please combine Figure 6 and Table 1. The description from line 162 – 167 is redundant.

5.     Please add error bars to the data in Table 1.

6.     Line 175 – 176, the statement is not comprehensible. Please revise it with clear language.

7.     Line 178 – 181, what are the raw materials that exfoliated from the wall of Geopolymer membrane? Line 181 – 182, as there were cracks for geo and geo-Tio2 membranes, then you could not compare these two membranes with others. The measurements were not performed at the same level. Line 182 – 185, the salt rejection was reported by you or He and Abukhadra?

8.     Line 199, typo: tensile splitting tensile.

9.     Line 204 – 208, please remove trivial details.

10.  FTIR characterization: 1. please specify which spectrum you were discussing. 2. Where are 594 and 445 cm-2 in the figure? Line 226 – 227, does 560 in (a) also due to Ti-O bonds? What is the difference between 561 and 560 you showed in the figure? In general, FTIR analysis was not acceptable. Please thoroughly revise this section.

11.  XRD: with and without rGO, there is no difference in XRD, so, what is the d-spacing of the rGO you used?

12.  SEM-EDS: 1. Without comparing with the control (membrane before the experiment), I think you cannot conclude any morphology differences. 2. Also, if there were indeed “morphological destruction” by high temperature and pressure, it implies that the membranes you used are not suitable for such operations. If this is the case, what is the point of this work? 3. By “looking at SEM figures”, I don’t see any TiO2 NP and rGO. You may need additional data to support this observation, such as EDX.

13.  Line 268 – 269, “Figure xxx”?

14.  Line 269 – 272: have you found out this conclusion or others?

15.  Figure 11 is for surface or cross-section? The figure description is not consistent with the Figure caption. Also, where are the corresponding SEM figures for the EDX mapping? Line TiO2 is for anti-fouling or biofouling?

Author Response

Thank you for your insightful comments and suggestions on the paper title: Pervaporation Membranes for Seawater Desalination Based on Geo-rGO-TiO2 Nanocomposites. Part 2. Membranes Performances. We have gone through all the comments critically and incorporated the changes in the revised manuscript accordingly.

Please refer to the list below to see the author’s response to the comments and suggestions provided.

Point 1: In materials and methods, could you please briefly introduce your synthesis process? If I don’t read your first work, I don’t understand what these membranes are. For example, the difference between Geo-rGO-TiO2 (0.5, 3) and Geo-rGO-TiO2 (1, 3).

Response 1: Thank you for your advice.

The synthesis procedure and membrane type are described briefly in the revised manuscript's section on materials and methods.

The geopolymer paste was made from metakaolin and activated with an alkaline solution (NaOH + Na2OSiO2 + H2O), then sealed in a closed mold and cured at 70 °C for 2 hours. The TiO2NPs used in this study were supplied by Sanno in Indonesia and had an average particle size of 20 nm. Hummer's modified method is used to create rGO nanosheets from graphite. The synthesis of geopolymer and geopolymer-rGO-TiO2 composites is carried out manually. Table 1 shows the four types of pervaporation membrane compositions developed in this study for seawater desalination. Each composition is made up of several specimens for mechanical testing, microstructure examinations, and pervaporation experiments. The amount of geopolymer paste in each membrane is the same. The variations of rGO addition are intended to improve the photocatalytic properties of TiO2NPs while increasing the salt rejection capacity of the formed composite.

Table 1. The composition of starting materials (in grams) of the pervaporation membranes produced in this study.

Sample

Metakaolin

NaOH

Na2OSiO3

H2O

rGO

TiO2

Geopolymer

30

3

24

9

0

0

Geo-TiO2(3)

30

3

24

9

0

3

Geo-rGO-TiO2 (0.5, 3)

30

3

24

9

0.5

3

Geo-rGO-TiO2 (1, 3)

30

3

24

9

1.0

3

Point 2: Line 145 – 158. First of all, if the experimental sample was physically affected by experimental conditions, then, you need to optimize the experimental conditions. Otherwise, your data was significantly compromised. For example, due to membrane cracking, you reduced the pressure. What about other membranes? Did you also reduce the pressure for other sets of experiments? Second of all, although you reduced the pressure, you observed membrane cracking in the end. Except for the variables (e.g., flux) you were interested in, you brought additional variables (i.e., materials strength) in the experiments, which invalidated your data in Figure 5. In this case, I could conclude the higher flux for Geopolymer and Geo-TiO2 was due to membrane cracking. Please optimize your experimental conditions to ensure all samples were measured at the same level.

Response 2: Thank you for this suggestion. The explanation for the experimental condition has been revised.

The flux permeate and membrane durability were tested in preliminary experiments at different pressures and temperatures, vacuum pressure, water cooling temperature, and pervaporation time. Following two trial experiments, it was discovered that the geopolymer and Geo-TiO2NPs membranes exhibited significant cracking.  Based on these findings, the pervaporation for all prepared membranes were conducted by adjusting the feed pressure to 2 psi, setting the feed temperatures to 40°, 60°, and 80 °C, a vacuum pressure of 0.75 bar, and a water-cooling temperature of 15 °C. Each set of experiments took about 5- 6 hours to complete in order to collect enough permeate for testing. Under the established pervaporation parameters, it was discovered that geopolymers and Geo-TiO2 (3) membranes still have significant cracks and thus reveal relatively high flux permeate, whereas Geo-rGO-TiO2 (0.5, 3) and Geo-rGO-TiO2 (1, 3) membranes had no surface cracks. As a result, the produced geopolymers and Geo-TiO2 (3) membranes are identified as being unable to withstand the feed's pressure and high temperature. The flux permeate of the membranes at various temperatures is depicted in Figure 5.

Point 3: Line 145 – 158, please reorganize your data set. After describing Figure 5, you may be followed by Figure 6 and so on. Also, can you please provide a figure or table to compare the flux from this work with others? or a brief description.

Response 3: Thank you very much for this suggestion. The whole data set has been reorganized as suggested. Table 4 shows comparison between the produced membrane performance and other inorganic membranes.

Point 4: Please combine Figure 6 and Table 1. The description from line 162 – 167 is redundant.

Response 4: Thank you for this suggestion. Figure 6 has been removed, and Table 1 has been updated. The description from lines 162–167 has been revised.

Point 5: Please add error bars to the data in Table 1.

Response 5: I appreciate your advice. The information in table 1 is split into two sets of data (table 3 in the updated manuscript). The data for flux and pH permeate were measured in our lab and have error bars based on their standard deviation, while the data for turbidity, salinity, TSS, and EC were measured in the lab of the Environmental Management Agency (EMA) in South Sulawesi Province, Indonesia, and initially they do not provide any error bars. The data in Table 3 has been added with error bars based on the instrument’s accuracy provided by the lab. operator of EMA as requested.

Point 6: Line 175 – 176, the statement is not comprehensible. Please revise it with clear language.

Response 6: Thank you for this comment. Table 3 shows that the salt rejection for the Geo-rGO-TiO2 (0.5, 3) and Geo-rGO-TiO2(1,3) membranes were around 73.01% and 91.26%, respectively. Although the salt rejection value of the two membranes did not reach 99%, the addition of rGO-TiO2NPs to the matrix geopolymer appears to play a significant role in separating NaCl from H2O. It is highly probable that the low salt rejection of Geo-rGO-TiO2 (0.5, 3) and Geo-rGO-TiO2 (1, 3) membranes is due to their average size of pore, which is 12.90 nm and 13.40 nm, respectively. In contrast to the NaA-Zeolite geopolymer membrane, which has an average pore size of about 3.77 nm and achieves 99% salt rejection [20], the Geo-rGO-TiO2 membranes generated in this study have substantially larger average pore sizes.

Point 7: Line 178 – 181, what are the raw materials that exfoliated from the wall of Geopolymer membrane? Line 181 – 182, as there were cracks for geo and Geo-TiO2 membranes, then you could not compare these two membranes with others. The measurements were not performed at the same level. Line 182 – 185, the salt rejection was reported by you or He and Abukhadra?

Response 7: Thank you for these questions. Our response has been added to the manuscript as follows. The measured TSS values of the membrane geopolymer and Geo-TiO2 (3) appear to be very high, exceeding the TSS value of seawater (feed). This value is unreasonable and is most likely due to the presence of alkali carbonate crystals detached from the surface and the cracks area of geopolymers and the Geo-TiO2 (3) membranes. One of the concerns with geopolymers made from metakaolin or fly ash is the formation of white needle-like alkali carbonate crystals known as efflorescence, which is caused by the mobility of excess alkali when the geopolymer is exposed to the open air or submerged in water for an extended period of time [26,27]. Despite the absence of surface cracks in the Geo-rGO-TiO2(0.5,1) and Geo-rGO-TiO2(1,5) membranes, NTU and TSS values are still high and detectable when NaCl rejection reaches 91%. Both membranes could also experience efflorescence, and in this case, the presence of high NTU and TSS values is due to the formation of alkaline carbonate crystals. More research is needed to confirm this.

Point 8: Line 199, typo: tensile splitting tensile.

Response 8: Thank you for this correction. The typo has been corrected.

Point 9: Line 204 – 208, please remove trivial details.

Response 9: Thank you for this suggestion. The trivial details have been removed.   

Point 10: FTIR characterization: 1. please specify which spectrum you were discussing. 2. Where are 594 and 445 cm-2 in the figure? Line 226 – 227, does 560 in (a) also due to Ti-O bonds? What is the difference between 561 and 560 you showed in the figure? In general, FTIR analysis was not acceptable. Please thoroughly revise this section.

Response 10: Thank you for this correction. The whole part of FTIR result and discussion has been revised as follow. Figure 6 shows the FTIR spectrum of the membranes after the pervaporation process was completed. The stretching and bending vibrations of the functional groups of (OH)- for membrane geopolymer are centered on the broad band at approximately 3448 cm-1 and 1644 cm-1, respectively. The centers of these broad bands are slightly shifted in other membranes, with the largest shift occurring in the Geo-rGO-TiO2 (1,3) membrane, specifically at positions 3464 cm-1 and 1649 cm-1. The bands that are identified as the primary functional groups of geopolymers are as follows: The asymmetrical stretching vibration of the Si-O-Si bond is observed at around 1014 cm-1, while the bending vibration of the Si-O-Si bond is discovered at around 456 cm-1. The band around 861 cm-1 is attributed to the stretching vibration of the Si–O bond, and the stretching vibration of the Al-O-Si bond is approximately 551 cm-1. The bending vibration of Al(IV)-O-Si in the cyclic structure of the geopolymer was found to be 687 cm-1 [31-33]. Except for the asymmetrical stretching vibration of the Si-O-Si bond, the addition of TiO2NPs and rGO to the geopolymer network caused a noticeable peak shift in the other primary groups. The functional group of Ti-O-Ti bonds in the membrane to which TiO2NPs are added vibrates at a wave number of approximately 1403 cm-1 [13]. The asymmetric stretching of the O-C-O bond within the Na2CO3 group is centered around 1412 cm-1 [35].

Figure 8. The FTIR spectrum of the membranes after a series of pervaporation experiments indicating the vibrational bands of the functional groups of (a) geopolymer, (b) Geo-TiO2 (3), (c) Geo-rGO-TiO2 (0.5, 3), and (d) Geo-rGO-TiO2 (1, 3).

Point 11: XRD: with and without rGO, there is no difference in XRD, so, what is the d-spacing of the rGO you used?

Response 11: Thank you for this comment. The XRD analysis of the samples has been confirmed and presented to demonstrate the effect of the addition of rGO. The diffractogram of rGO used in this study, as reported in our first article, is as follows. The reflection peak of rGO with an index Miller (002) appears at 2q = 26.00o with d-spacing = 3.424 angstroms (PDF#01-077-7164).

The diffractogram of the membranes is shown as follows. The addition of rGO alters the geopolymer's amorphous network by shifting the hump peak to a smaller angle of 2q. The reflection peak of TiO2NPs with a crystallographic plane of (101) is at 2q = 25.57o and d-spacing = 3.4807 angstroms (PDF#01-075-2552). It appears that the reflection angle of crystallographic plane TiO2NPs (101) overlaps with the reflection angle of rGO (002).

Point 12: SEM-EDS: 1. Without comparing with the control (membrane before the experiment), I think you cannot conclude any morphology differences. 2. Also, if there were indeed “morphological destruction” by high temperature and pressure, it implies that the membranes you used are not suitable for such operations. If this is the case, what is the point of this work? 3. By “looking at SEM figures”, I don’t see any TiO2 NP and rGO. You may need additional data to support this observation, such as EDX.

Response 12: Thank you for your important points on the SEM data. Our response is as follows.

The SEM images of all the as-prepared membranes have been reported in our first article (Membranes, 2021,11, 966). The SEM images show the morphology of the membranes before pervaporation. A close examination of the membrane surface morphology before and after pervaporation reveals that there is morphological destruction caused by the feed pressure and temperature. We did not include these SEM images in the second manuscript to avoid using the same SEM images. In addition, the SEM images unfortunately were taken at different machine, since this instrument is not available in our laboratory.

(a)

(b)

(c)

(d)

This discovery is significant because it provides new insights into improving the quality of geopolymers, a material that has recently been extensively researched for use in waste water treatment and seawater desalination. The quality of the geopolymer, an inorganic substance synthesized by the alkali activation method, depends on a number of variables, including the type and particle size of the starting material, the type and atomic molar ratio of the alkali species (Si/Al, Na/Al, and Na/H2O), curing time, and temperature. It is quite challenging to manufacture geopolymer as a membrane layer free of defects.

Furthermore, Figure xxx depicts the distribution of rGO particles and TiO2NPs for the morphology of the membrane surface using x-ray mapping.

Point 13: Line 268 – 269, “Figure xxx”?

Response 13: Thank you for this correction. The figure number has been corrected to be Figure 4.

Point 14: Line 269 – 272: have you found out this conclusion or others?

Response 14: Thank you for this question. The statements in lines 269-272 were made based on our experimental results from a series of tests on Geo-rGO-TiO2 membranes. The results showed that incorporating rGO into the geopolymer paste has a significant impact on preventing cracks on the surface of the membranes. This finding is supported by other reports [18,38,39].

Point 15: Figure 11 is for surface or cross-section? The figure description is not consistent with the Figure caption. Also, where are the corresponding SEM figures for the EDX mapping? Line TiO2 is for anti-fouling or biofouling?

Response 15: Thank you very much for correcting the caption to Figure 9. The image has been corrected to show that it is an X-Ray Mapping of the elemental distribution on the surface of the membranes. The corresponding SEM images for the X-Ray Mapping have also been included.

For line 284, the statement of TiO2NPs as anti-fouling has been changed to biofouling. According to Mario Andreossi et al. [18], the rGO/TiO2 nanocomposite has hydrophobic natural properties derived from phase carbonaceous, confirming its potential as an efficient biofouling material, particularly for removing hydrophobic organics, i.e., aromatics.

Reviewer 2 Report

The authors intended to develop nanocomposite membranes for seawater desalination, but obviously the membranes developed are of poor quality. The salt rejection of the membranes is much lower than the commercial standard (99%) and its water flux is also EXTREMELY low.

Authors discussed a lot on the membrane surface properties (published in Part 1’s paper) and the fouling, but detailed discussion on why their developed membrane has poor rejection is not given. A dense membrane for pervaporation process should achieve almost complete salt rejection. Obviously, the membrane developed in this work has severe surface defects.

Some of the data presented by the authors are also contradictory. For instance, in Table 1, Geo-rGO-TiO2 (1, 3) membrane achieved the highest salt rejection, but its removal rate of turbidity is the lowest. Besides, I have no idea why total suspended solids (TSS) could still be detected in the permeate of this sample. At this NaCl rejection (>91%), both TSS and NTU should be EXTRMELY low or not detected.

There are many issues associated with the manuscript.

Introduction – There is no literature at all about why nanocomposite membrane could be a better candidate for seawater desalination. Also, on what perspective the pervaporation could compete with TFC RO membrane for desalination process?

Figure 1 – What is the point of showing this figure in Introduction? This is a technical paper, NOT review article.

Methodology – Authors should provide a summary table on the key properties of the membranes used in this work (based on the results published (part 1).

The temperature of cooling water should be reported.

Long-term stability of the membrane should be provided (at least 48 hours).

Results and discussion – Figure 6 – I hardly see any difference between these water samples!

Figure 7 – It is quite obvious that membrane suffered from severe fouling within short period of operation. The surface colour changed by comparing with Figure 3. This contradicts to TiO2 and GO which are hydrophilic materials that can improve antifouling.

A table to compare the performance of developed membranes with other literature data should be provided.

Author Response

We appreciate your thorough review of our manuscript.

First of all. The primary objective of this research is to develop a pervaporation membrane for seawater desalination based on geopolymer and its composite incorporating rGO and TiO2NPs. Unlike other inorganic or organic materials that have been more maturely developed as pervaporation membranes, the fabrication of pure geopolymer or its composite as the raw material of membranes is still in its early stages, and has not been widely studied. The quality of the geopolymer, an inorganic substance synthesized by the alkali activation method, depends on a number of variables, including the type and particle size of the starting material, the type and atomic molar ratio of the alkali species (Si/Al, Na/Al, and Na/H2O), curing time, and temperature. It is quite challenging to manufacture geopolymer as a membrane layer free of defects. We are aware that the quality of the produced membranes is yet in no way up to commercial standards, and this gives us with a more extensive study opportunity.

Second. The potential causes of the low salt rejection of the generated membranes were discussed in the updated manuscript. It is highly probable that the low salt rejection of Geo-rGO-TiO2 (0.5, 3) and Geo-rGO-TiO2 (1, 3) membranes is due to their average size of pore, which is 12.90 nm and 13.40 nm, respectively. In contrast to the NaA-Zeolite geopolymer membrane, which has an average pore size of about 3.77 nm and achieves 99% salt rejection [20], the Geo-rGO-TiO2 membranes produced in this study have substantially larger average pore sizes.

Third. The data shown in table 1 (which is amended to table 2 in the revised version), including turbidity, TSS, EC, and salinity, were measured in the Environmental Management Agency laboratory in South Sulawesi Province, Indonesia, and their measurement results were reported as we received. In the results and discussion section, some potential reasons for the low turbidity the Geo-rGO-TiO2 membrane (1,3) are discussed.

One of the concerns with geopolymers made from metakaolin or fly ash is the formation of white needle-like alkali carbonate crystals known as efflorescence, which is caused by the mobility of excess alkali when the geopolymer is exposed to the open air or submerged in water for an extended period of time [26,27]. Despite the absence of surface cracks in the Geo-rGO-TiO2(0.5,1) and Geo-rGO-TiO2(1,5) membranes, NTU and TSS values are still high and detectable when NaCl rejection reaches 91%. Both membranes could also experience efflorescence, and in this case, the presence of high NTU and TSS values is due to the formation of alkaline carbonate crystals. The efflorescence in geopolymer made from metakaolin can be controlled by adjusting the molar rasio of Na2O/Al2O3 of the alkali solution used [28], or the addition of soluble silicate, silica fume or silicone oil [29]. It is unclear whether the addition of rGO will be able to control the formation of efflorescence on the surface of the geopolymer. More research is needed to confirm this.

Point 1: Introduction – There is no literature at all about why nanocomposite membrane could be a better candidate for seawater desalination. Also, on what perspective the pervaporation could compete with TFC RO membrane for desalination process?

Response 1: Thank you for this important point. The revised manuscript also incorporates some relevant literature reviews on nanocomposite PV and TFC RO.

The fabrication of thin film composite (TFC), which is composed of m-phenylenediamine (MPD) and trimesoyl chloride (TMC) on a porous substrate, has been the primary method for manufacturing reverse osmosis (RO) membranes. The TFC RO membrane exhibits remarkable performance in producing pure water from seawater with high flux and salt rejection and good mechanical strength. However, it has been discovered that TFC RO membranes are prone to fouling and influence the cost of desalination. The development of thin film nanocomposite (TFN) RO membranes is one concept that might be used to address the drawbacks of TFC RO. This form of membrane combines high hydrophilicity, anti-fouling, and anti-microbial qualities with the functional properties of nanoparticles [13,14]. Currently, another alternative to enhance the performance of nanocomposite membranes is to fabricate membranes using the layer-by-layer (LBL) method by integrating different kinds of nanomaterials with polymer polyelectrolytes [15].

In recent years, a number of studies have reported the incorporation of nanomaterial into a polymer to fabricate nanocomposite PV membranes for seawater desalination, such as graphene oxide (GO) [16], chitosan/GO [6], AgNPs [17], rGO-TiO2 [18], Al2O3 NPs [19]. The next generation of high-performance membranes for PV desalination could be developed using nanocomposite membranes, which have the potential to compete with established RO membranes and overcome the shortcomings of low permeability and salt rejection, improved hydrophilicity and mechanical strength, and enhanced antifouling and antibacterial properties.

Point 2: Figure 1 – What is the point of showing this figure in Introduction? This is a technical paper, NOT review article.

Response 2: Thank you for this comment.

We agree with the reviewer. As a technical paper, Figure 1 is not very relevant to be listed and has therefore been removed. 

Point 3: Methodology – Authors should provide a summary table on the key properties of the membranes used in this work (based on the results published (part 1).

Response 3: Thank you for this suggestion. A summary table of the key properties of the membranes produced in this study has been added and explained.

Point 4: The temperature of cooling water should be reported.

Response 4: Thank you for this suggestion. The cooling water temperature during the pervaporation was around 15 °C and it has been added in the manuscript.

Point 5: Long-term stability of the membrane should be provided (at least 48 hours).

Response 5: Thank you for making this suggestion. This is a critical point to make. Unfortunately, running experiments for at least 48 hours without a break is not possible in our laboratory due to strict university regulations that prohibit us from running any experiments after 6 p.m. or working overnight at the laboratory. This regulation was enacted for security reasons and has been in effect for more than three years. In accordance with regulations, the pervaporation experiment is run for a maximum of 5-6 hours per day, followed by the next day.

To determine the membranes' long-term durability, the pervaporation time required to produce a 1 liter volume of permeate was used as the standard. This volume of permeate is required by the Environmental Management Agency in South Sulawesi Province, Indonesia, to measure turbidity, total suspended solids (TSS), salinity, and electrical conductivity (EC).

Table 4. Pervaporation time to produce 1 liter permeate.

Membrane

Flux (kg/m2h)

Pervaporation time (h)

Geopolymer

5.29

47.5

Geo-TiO2 (3)

8.06

31.2

Geo-rGO-TiO2 (0.5, 3)

0.92

273

Geo-rGO-TiO2 (1, 3)

2.29

109

Point 6: Results and discussion – Figure 6 – I hardly see any difference between these water samples!

Response 6: Thank you for this statement. Figure 6 was omitted because it was unclear.

Point 7: Figure 7 – It is quite obvious that membrane suffered from severe fouling within short period of operation. The surface colour changed by comparing with Figure 3. This contradicts to TiO2 and GO which are hydrophilic materials that can improve antifouling.

Response 7: Thank you for suggesting this. True, the membrane experienced severe fouling in a short period of time. The corrosion inside the water pump we used was the main source of fouling, and it spread along the pipe line to the membrane house. During the course of this study, we did not have the opportunity to replace the water pump. However, we discovered that after the pervaporation process is complete, the fouling from the membrane's surface can be cleaned with running water, indicating that the self-cleaning properties of TiO2NPs are effective.

Point 8: A table to compare the performance of developed membranes with other literature data should be provided.

Response 8: Thank you for this suggestion. A comparison of the performance of the produced membrane with other inorganic membranes has been included.

Table 4. Comparison between the produced membrane performance and other inorganic membranes.

Membrane

Thickness (mm)

Feed temperature (°C)

Flux (kg/m2.h)

Rejection (%)

References

NaZ zeolite-geopolymer

10

90

3.86

99.5

[20]

Zeolitized diatomite/geopolymer

3

90

7.63

97.6

[21]

Hydroxy sodalite

2

202

3.5

99.99

[30]

Geo-rGO-TiO2 (1, 3)

2

60

2.92

91.26

This study

Reviewer 3 Report

1.  There are other geopolymer-based pervaporation membranes having been reported for desalination purposes. The authors may want to include some relevant studies in the introduction section. 

2. Equations (1) and (2): The symbol of multiplication should be "´" instead of .

3. What is the temperature of the circulating cooling reagent?

4. Did the author test the TOC level of the permeate water?

5. How long did each pervaporation experiment last?

6. With cracks, 

7. With rGO, the permeate flux dropped quite significantly. Can the authors explain why Geo-rGO-TiO2 (1, 3) achieved higher flux than Geo-rGO-TiO2 (0.5, 3)?

8. Table 1 is hard to read. Please indicate clearly what items are for the feed and what items are for the permeate. From the EC levels, the permeate contains significant amount of salts. This is not consistent with the salinity. The poor permeate quality discloses that either the membranes have defects or large pores.

9. Section 3.2: "The cracks could have been caused by fairly high pressure..." This explanation is not convincing as the maximum operating water pressure is only 2 psi.

10. Can the authors explain why high temperature causes decreased membrane strength?

11. The membrane surface was contaminated during experiment, and the authors claimed that this was due to corrosion in the pump. Did the authors do any check or test or characterization to confirm this?

Author Response

Thank you for your insightful comments and suggestions on the paper titled: Pervaporation Membranes for Seawater Desalination Based on Geo-rGO-TiO2 Nanocomposites. Part 2. Membranes Performances. We have gone through all the comments critically and incorporated the changes in the revised manuscript accordingly.

Please refer to the list below to see the author’s response to the comments and suggestions provided.

Point 1: There are other geopolymer-based pervaporation membranes having been reported for desalination purposes. The authors may want to include some relevant studies in the introduction section.

Response 1: Thank you for this suggestion. Pervaporation membranes for seawater desalination based on geopolymer has been included in introduction as follows.

He et al. [20] reported on the development of geopoymer-based material for seawater desalination. The Al2O3-2SiO2-Na2O geopolymer membrane used by the authors was synthesized by a gel casting procedure, and it was afterwards hydrothermally converted into a NaA zeolite. The generated membrane, which has a thickness of 10 mm and a feed temperature of 90 °C, achieves a flux of 3.86 kg.m-2.h-1 and a salt rejection of about 99%.

A zeolitized geopolymer-based pervaporation membrane was developed and reported recently by Abukhadra et al. [21]. Natural diatomite and kaolinite were used to synthesize the geopolymer at room temperature. A zeolitized geopolymer membrane was then generated by hydrothermal zeolite growth at 100 °C for 24 hours. The authors examined the influence of the membrane thickness and feed temperature on the flux and salt rejection capacity of the membrane. Their results showed that for the tested membrane at a thickness of 3 mm, and feed temperature of 90 °C, the water flux reached values of 7.05 kg.m-2.h-1 with a salt rejection of around 99,57% after 130 h of pervaporation time.

Point 2: Equations (1) and (2): The symbol of multiplication should be "´" instead of

Response 2: Thank you for this correction. The symbol of multiplication in equation (1) and (2) has been changed. 

Point 3: What is the temperature of the circulating cooling reagent?

Response 3: The minimum temperature of cooling water is 15 °C.

Point 4: Did the author test the TOC level of the permeate water?

Response 4: Thank you for this question. Unfortunately, the measurement of TOC was not available at the Environmental Management Agency in South Sulawesi Province, Indonesia. The machine for measuring TOC levels in our lab is currently out of order.

Point 5: How long did each pervaporation experiment last?

Response 5: Thank you for this question. The pervaporation experiment for all membranes was performed every day between 5 to 6 hours until we got 1 liter of permeate. Table 4 shows the pervaporation time needed to collect 1 liter of permeate.

Point 6: With cracks,

Response 6: This point is presumably connected to inquiry number 5. Until both membranes were removed from the membrane house once the level of permeate reached 1 liter, we were unaware that cracks were developing on the Geopolymer and Geo-TiO2 (3) membranes.

Point 7: With rGO, the permeate flux dropped quite significantly. Can the authors explain why Geo-rGO-TiO2 (1, 3) achieved higher flux than Geo-rGO-TiO2 (0.5, 3)?

Response 7: Thank for this important question.

We described the physical characteristics of the generated membranes in our first article (membranes 2021, 11, and 966) Geo-rGO-TiO2 (0.5) has a thickness and density of 2.3 mm and 0.98 g.cm-3, respectively. Geo-rGO-TiO2 (1, 3) has a thickness and density of 2.2 mm and 0.97 g.cm-3. Additionally, based on BET measurements, it was discovered that the average pore size of the Geo-rGO-TiO2(0.5, 3) membrane is approximately 12.90 nm, while the average pore size of the Geo-rGO-TiO2 (1, 3) membrane is approximately 13.40 nm. The main causes of the difference in permeate flux between Geo-rGO-TiO2 (1, 3) and Geo-rGO-TiO2 (0.5, 3) are variations in membrane thickness, density, and pore size.

Point 8: Table 1 is hard to read. Please indicate clearly what items are for the feed and what items are for the permeate. From the EC levels, the permeate contains significant amount of salts. This is not consistent with the salinity. The poor permeate quality discloses that either the membranes have defects or large pores.

Response 8: I appreciate you making this observation. Table 1 has been changed into Table 2, which is coupled with Figure 5 to make it simpler to understand.

Point 9: Section 3.2: "The cracks could have been caused by fairly high pressure..." This explanation is not convincing as the maximum operating water pressure is only 2 psi.

Response 9: Thank you for this comment.

Geopolymer is a cementitious material known to have good compressive strength, but it is inherently brittle and lacks adequate ductility. The brittleness of geopolymer makes geopolymer structural elements susceptible to cracking under certain pressure or tensile. In order to improve this drawback, geopolymer paste is normally reinforced with natural or synthetic fibers such as carbon fiber, nanoparticles such as nano quartz, or graphene-based materials such as carbon nano tubes (CNTs) and nano graphene oxide (NGO), which can be worked to control crack propagation, enhance toughness, and deformation as well as improve the microstructure of geopolymer [1]–[3].

In our experiment, it was discovered that the Geo-rGO-TiO2NPS membranes were intact even after a lengthy working pervaporation time. Geopolymer membranes and Geo-TiO2(3), in contrast, show considerable cracks in a much shorter pervaporation time. This result reveals that the geopolymer paste surface cracks even at water pressures of roughly 2 psi, and that the cracking becomes worse at high feed temperatures.

[1]         A. G. N. Abbas, F. N. A. A. Aziz, K. Abdan, N. A. M. Nasir, and G. F. Huseien, “A state-of-the-art review on fibre-reinforced geopolymer composites,” Constr. Build. Mater., vol. 330, no. March, p. 127187, 2022.

[2]         H. U. Ahmed, A. A. Mohammed, and A. S. Mohammed, “The role of nanomaterials in geopolymer concrete composites: A state-of-the-art review,” J. Build. Eng., vol. 49, no. January, p. 104062, 2022.

[3]         M. M. Mokhtar, S. A. Abo-El-Enein, M. Y. Hassaan, M. S. Morsy, and M. H. Khalil, “Mechanical performance, pore structure and micro-structural characteristics of graphene oxide nano platelets reinforced cement,” Constr. Build. Mater., vol. 138, pp. 333–339, 2017.

Point 10: Can the authors explain why high temperature causes decreased membrane strength?

Response 10: Thank you very much for this question.

Geopolymer is a heat- and temperature-resistant inorganic polymer that is widely used in construction. Even at 1000 oC, this material does not burn, although having significant cracks [4], [5]. The internal stress that the temperature will put on the geopolymer's structural element will cause cracks to start growing, as is well known. The strength of the pure geopolymer membrane will decrease as cracks develop. Adding graphene-based components like GO or rGO to geopolymers is one method used to increase their thermal strength [6].

[4]         D. dos S. da Silva Godinho, F. Pelisser, and A. M. Bernardin, “High temperature performance of geopolymers as a function of the Si/Al ratio and alkaline media,” Mater. Lett., vol. 311, no. August 2021, p. 131625, 2022.

[5]         Subaer and A. Van Riessen, “Thermo-mechanical and microstructural characterization of sodium-poly (sialate-siloxo)(Na-PSS) geopolymers,” J. Mater. Sci., vol. 42, pp. 3117–3123, 2007.

[6]         M. Li et al., “High temperature properties of graphene oxide modified metakaolin based geopolymer paste,” Cem. Concr. Compos., vol. 125, no. October 2021, p. 104318, 2022.

Point 11: The membrane surface was contaminated during experiment, and the authors claimed that this was due to corrosion in the pump. Did the authors do any check or test or characterization to confirm this?

Response 11: Thank you for this comment. The corrosion inside the water pump is very obvious and we provide a photograph of it.

The figure shows the accumulation of corrosion materials in the part of the water pump.

Round 2

Reviewer 1 Report

I thank the authors for addressing my comments. I would like to recommend accepting it for publication.

Reviewer 2 Report

I'm satisfied with the actions taken by the authors.